# Analysis of Early Biomarkers Associated with the Development of Critical Respiratory Failure in Coronavirus Disease 2019 (COVID-19)

**DOI:** 10.3390/diagnostics12020339

**Published:** 2022-01-28

**Authors:** Hiroyoshi Yamada, Masaki Okamoto, Yoji Nagasaki, Suzuyo Yoshio, Takashi Nouno, Chiyo Yano, Tomohiro Tanaka, Fumi Watanabe, Natsuko Shibata, Yoko Arimizu, Yukako Fukamachi, Yoshiaki Zaizen, Naoki Hamada, Atsushi Kawaguchi, Tomoaki Hoshino, Shigeki Morita

**Affiliations:** 1Department of Respirology and Clinical Research Center, National Hospital Organization Kyushu Medical Center, 1-8-1 Jigyohama, Chuo-ku, Fukuoka 810-0065, Japan; 0636.h.y@gmail.com (H.Y.); nouno_takashi@med.kurume-u.ac.jp (T.N.); yano_chiyo@med.kurume-u.ac.jp (C.Y.); tanaka_tomohiro@med.kurume-u.ac.jp (T.T.); fuchan.2408@gmail.com (F.W.); shibata.natsuko.pk@mail.hosp.go.jp (N.S.); 2Division of Respirology, Neurology and Rheumatology, Department of Internal Medicine, Kurume University School of Medicine, 67 Asahi-machi, Fukuoka 830-0011, Japan; zaizen_yoshiaki@med.kurume-u.ac.jp (Y.Z.); hoshino@med.kurume-u.ac.jp (T.H.); 3Department of Infectious Disease, National Hospital Organization Kyushu Medical Center, 1-8-1 Jigyohama, Chuo-ku, Fukuoka 810-0065, Japan; nagasaki.yoji.up@mail.hosp.go.jp (Y.N.); yoko.nakashima1021@gmail.com (Y.A.); yukako.f28@gmail.com (Y.F.); 4Shino-Test Corporation, 3-7-9 Kanda Surugadai, Chiyoda-ku, Tokyo 101-8410, Japan; suzuyo.shibata@shino-test.co.jp; 5Research Institute for Diseases of the Chest, Graduate School of Medical Sciences, Kyushu University, 3-1 Maidashi Higashi-ku, Fukuoka 812-0054, Japan; hamada.naoki.608@m.kyushu-u.ac.jp; 6Education and Research Center for Community Medicine, Faculty of Medicine, Saga University, Saga 849-8501, Japan; akawa@cc.saga-u.ac.jp; 7Department of Cardiovascular Surgery, National Hospital Organization Kyushu Medical Center, 1-8-1 Jigyohama, Chuo-ku, Fukuoka 810-0065, Japan; morita.shigeki.mu@mail.hosp.go.jp

**Keywords:** coronavirus disease, biomarker, Krebs von den Lungen-6, high-mobility group box-1 protein, interleukin-6

## Abstract

Certain biomarkers predict death due to acute respiratory distress syndrome in COVID-19 patients. We retrospectively analyzed biomarkers associated with time to mechanical ventilation for respiratory failure due to COVID-19 (time-to-mechanical ventilation) in 135 consecutive patients in our hospital. We analyzed biomarkers that were elevated immediately (at admission) and later (3 days after admission) using Cox proportional hazards regression analysis. Independent biomarkers of time-to-mechanical ventilation were high C-reactive protein (CRP), interleukin (IL)-6, and Krebs von den Lungen-6 (KL-6) concentrations at admission and elevated CRP, high-mobility group box-1 protein (HMGB-1), and d-dimer levels and low platelets 3 days after admission. Receiver operating characteristic analysis for detecting the association between independent biomarkers associated with time-to-event in multivariate analyses and the start of mechanical ventilation revealed that these biomarkers had area under the curve values higher than 0.700. The present study suggests that CRP was the only biomarker associated with time-to-mechanical ventilation both at admission and 3 days after admission. Moreover, IL-6 (an inflammatory cytokine), HMGB-1 (a late inflammatory mediator), and KL-6 (reflecting injury and/or remodeling of type II pneumocytes) were associated with outcomes in COVID-19 as reported previously. In conclusion, increased CRP, IL-6, KL-6, HMGB-1, and d-dimer levels and decreased platelet counts were associated with the start of mechanical ventilation due to COVID-19.

## 1. Introduction

The viral pneumonia outbreak that began in Wuhan, China, in December 2019 is presently ongoing, and the novel pathogen linked to this disease has been named severe acute respiratory syndrome coronavirus 2 (SARS-CoV-2). Pneumonia caused by this virus, named coronavirus disease 2019 (COVID-19), has spread rapidly worldwide [1,2]. Although most COVID-19 patients have bilateral pneumonia without hypoxemia and asymptomatic or mild disease, some patients develop lethal respiratory failure.

The main cause of death in COVID-19 is the development of acute lung injury/acute respiratory distress syndrome (ALI/ARDS) [1,2,3,4]. A systematic review and meta-analysis reported that the overall pooled mortality estimate among 10,815 ARDS cases in COVID-19 patients was 39% (95% confidence interval (CI): 23–56%) [4,5]. Mechanical ventilation (MV) is the key management strategy for ALI/ARDS in general infectious disease units (IDUs) or intensive care units (ICUs). Increased COVID-19 cases requiring MV can strain medical care resources in clinical practice. However, predicting the development of critical respiratory failure and stratifying patients early after admission may reduce this risk. Although some predictive blood biomarkers and risk factors, including older age, smoking, obesity, hypertension, diabetes mellitus, and coronary heart disease, have been reported, predictors of critical respiratory failure have not yet been established [3,4,5].

This study focused on determining biomarkers associated with the start of MV measured early after admission for COVID-19. An excessive immune response against SARS-CoV-2 induced by inflammatory cytokines and chemokines (i.e., cytokine storm) can contribute to the development of critical respiratory failure due to ALI/ARDS [6]. Previous studies have shown that serum levels of certain biomarkers were associated with disease severity and outcomes [7,8,9,10,11,12,13,14,15,16]. Moreover, studies using prediction models (ISARIC4C and COVID-GRAM) reported that the areas under the curve (AUC) for predicting the outcome of COVID-19 patients, including mortality and/or ICU admission, were 0.774 and 0.706, respectively [17,18].

In the present study, we attempted to clarify whether blood biomarkers measured early after admission are associated with the start of MV or death due to COVID-19.

## 2. Materials and Methods

### 2.1. Patients

This single-center retrospective study conducted a medical record review. The study subjects included 135 consecutive patients with COVID-19 who were admitted to our IDU from February 2020 to August 2020. The admission criterion to the IDU was a diagnosis of COVID-19 by SARS-CoV-2 real-time polymerase chain reaction using an oropharyngeal swab or sputum sample. Chest X-rays and computed tomography were performed in all patients to evaluate the presence of lower respiratory disease due to COVID-19. We classified the disease severity in patients with COVID-19 according to the National Institutes of Health criteria as follows [19]. Patients with COVID-19 with no symptoms consistent with COVID-19 were classified as the asymptomatic infection group. Patients with any of the various signs and symptoms of COVID-19 (e.g., fever, cough, sore throat, malaise, headache, muscle pain, nausea, vomiting, diarrhea, loss of taste, loss of smell) but without shortness of breath, dyspnea, or abnormal chest imaging were classified as the mild illness group. Patients who showed evidence of lower respiratory disease during clinical assessment or imaging and had >94% oxygen saturation (SpO_2_) in room air were classified as the moderate illness group. Patients with an SpO_2_ < 94% in room air were classified as the severe illness group, and patients with ARDS, septic shock, and/or multiple organ dysfunction were classified as the critical illness group. This classification of severity was determined by the condition at the time of admission.

### 2.2. Endpoints

The aim of the present study was to determine biomarkers associated with the start of MV in COVID-19 patients. Time-to-MV was defined as the time from admission to the start of MV to control respiratory failure due to COVID-19. The indication for MV was set similarly to that in clinical practice (i.e., respiratory failure that cannot be controlled by high-concentration oxygen therapy) [20].

### 2.3. Biomarker Analysis

We analyzed biomarker concentrations at admission and 3 days after admission to identify biomarkers elevated immediately and later after the onset of COVID-19. The serum concentrations of high-mobility group box-1 protein (HMGB-1), interleukin-6 (IL-6), IL-18 (an inflammasome-related cytokine) [21], and soluble CD163 (an M2-like macrophage-related biomarker shed from the cell surface by an inflammatory stimulus) were analyzed using residual serum [22]. The serum HMGB-1 level was measured using the HMGB1 ELISA Kit II (Shino-Test Co., Ltd., Sagamihara, Japan). The minimal detectable concentration (MDC) and coefficient of variation (CV) of this ELISA kit are 1 ng/mL and <10%, respectively. The serum level of IL-6 was measured by a chemiluminescent enzyme immunoassay using the human IL-6 measurement cartridge (MDC, 0.2 pg/mL; CV, 2.14–2.66%; Fujirebio Diagnostic Inc., Tokyo, Japan). The serum level of the inflammatory cytokine IL-18 was measured using a human IL-18 ELISA kit (MDC, 12.5 pg/mL; CV, 4.93–10.80%; Medical & Biological Laboratories Co., Ltd., Nagoya, Japan). The serum level of soluble CD163 was measured using a soluble CD163 ELISA kit (MDC, 0.177 ng/mL; CV, 3.4–3.8%; R&D Systems, Inc., Minneapolis, MN, USA). Other biomarkers including neutrophil, lymphocyte, and platelet counts and serum levels of C-reactive protein (CRP), lactate dehydrogenase (LDH), ferritin, Krebs von den Lungen-6 (KL-6), and d-dimer were measured as part of routine laboratory testing. Standard values of biomarkers were as follows: CRP, less than 0.20 mg/dL; KL-6, less than 500 IU/mL; LDH, less than 225 IU/L; IL-6, less than 6.6 pg/mL; soluble CD163, less than 472 pg/mL; neutrophil count, from 1800 to 7500/uL; lymphocyte count, 1000 to 4800/uL; platelet count range, 15.8 to 34.8 × 10^4^/uL.

### 2.4. Statistical Analysis

Data were expressed as the median (25th–75th percentiles of the interquartile range). Differences between multiple groups were analyzed as appropriate using the Wilcoxon rank-sum test for two groups and the Kruskal–Wallis test for three groups, or Fisher’s exact test. In Cox proportional hazards regression analysis, we detected variables significantly (*p* < 0.05) associated with time-to-MV by univariate analyses. Next, all significant variables detected by univariate analyses were analyzed by multivariate analysis using the backward elimination method. The biomarker cut-off levels were defined as the optimal value with the highest Youden index on receiver operating characteristic (ROC) curves generated using logistic regression, as in our previous report [23]. The validity of the logistic regression analysis result was verified by the Hosmer and Lemeshow goodness of fit (GOF) test. The validity of ROC analysis was verified by implementing 5-repeated 10-fold cross-validation. A value of *p* < 0.05 was considered to represent statistical significance, and all statistical analyses were performed using the JMP 14.0 program (SAS Institute Japan, Tokyo, Japan).

## 3. Results

### 3.1. Patient Characteristics and Outcomes

The patients’ characteristics are presented in Table 1. The study cohort consisted of 135 patients (69 males; median age, 50.0 years) with COVID-19, including 42 with asymptomatic infections or mild illness (mild illness group), 66 with moderate illness (moderate illness group), 27 with severe illness (severe illness group), and none with critical illness (critical illness group), in accordance with the severity at admission. Among the 135 study cases, 22 (16.3%) had diabetes mellitus (DM), 33 (24.4%) had hypertension, and 9 (6.7%) had malignant disease. A comparison of clinical data by the severity at admission showed that more severe disease was associated with older age and male gender. The comparison of biomarker levels at admission showed that more severe disease was significantly associated with higher neutrophil counts, CRP, LDH, ferritin, IL-6, KL-6, and d-dimer and lower lymphocyte counts. The comparison of biomarker concentrations 3 days after admission showed that more severe disease was significantly associated with higher neutrophil counts, CRP, LDH, ferritin, IL-6, IL-18, KL-6, HMGB-1, soluble CD163, and d-dimer and lower lymphocyte counts. There was no difference regarding smoking status, platelet counts at admission and 3 days after admission, or the levels of HMGB-1 and soluble CD163 at admission among the three severity groups.

The patients’ clinical courses are presented in Table 2. Among all 135 patients, 5 (3.8%) patients died of COVID-19 in hospital 10, 17, 22, 44, and 76 days after admission. The median hospital stay period was 9.0 (6.0–15.0) days. Ten (7.4%) patients received MV, and one (0.74%) patient died of COVID-19 without receiving MV. Among the 10 (7.4%) patients receiving MV, 6 (4.4%) survived, and 4 (3.0%) died in hospital. One patient who refused MV because of advanced age received high-concentration oxygen therapy by a reservoir mask and maximum pharmacological therapy and died 17 days after admission. 

### 3.2. Comparison of Biomarkers between Patients with and without MW

Table 3 shows the comparison of biomarker levels between cases with and without MV. Among the biomarkers at admission, patients with MV had significantly higher neutrophil counts, CRP, LDH, ferritin, and d-dimer and lower lymphocyte counts than those without MV. Similarly, patients with MV tended to have higher IL-6 and KL-6 levels and lower platelet counts than those without. IL-18, HMGB-1, and soluble CD163 levels were not significantly different between the two groups. Among the biomarkers 3 days after admission, patients with MV had significantly higher neutrophil counts, CRP, LDH, ferritin, IL-6, HMGB-1, and d-dimer and lower lymphocyte and platelet counts than those without MV or patients who survived. Soluble CD163 levels were not significantly different between the two groups. We could not measure KL-6 and IL-18 levels 3 days after admission owing to a lack of residual serum.

### 3.3. Cox Proportional Hazards Regression Analysis for Detecting Biomarkers Associated with Time-to-MV

The results of the univariate analysis are shown in Table 4. Among the biomarkers at admission, neutrophil counts and CRP, LDH, ferritin, IL-6, KL-6, and d-dimer levels were significantly associated with time-to-MV. Among the biomarkers 3 days after admission, neutrophil and platelet counts and CRP, LDH, ferritin, IL-6, HMGB-1, and d-dimer levels were significantly associated with time-to-MV.

The results of multivariate analyses using the backward elimination method are shown in Table 5. Among the biomarkers at admission, the independent biomarkers associated with time-to-MV were high levels of CRP (relative risk (RR), 33.1; 95% CI, 3.2–372.6; *p* = 0.0028), IL-6 (RR, 14.5; 95% CI, 0.67–142.7; *p* = 0.041), and KL-6 (RR, 64.7; 95% CI, 3.6–804.4; *p* = 0.0013). Among the biomarkers 3 days after admission, the independent biomarkers associated with time-to-MV were high levels of CRP (parameter estimate (PE) of RR, −0.33; PE of 95% CI, −0.96 to −0.044; *p* = 0.021), HMGB-1 (PE of RR, 0.22; PE of 95% CI, 0.054 to 0.46; *p* = 0.011), and d-dimer (PE of RR, 0.93; PE of 95% CI, 0.083 to 2.1; *p* = 0.00037) and low platelet counts (PE of RR, −0.37; PE of 95% CI, −0.72 to −0.14; *p* = 0.00060). The Hosmer–Lemeshow GOF test demonstrated the validity of the Cox proportional hazards regression analysis results of biomarkers at admission (*p* = 1.0) and 3 days after admission (*p* = 0.998) for predicting time-to-MV.

### 3.4. ROC Curve Analysis for Determining the Association between Biomarkers and the Start of MV

The results of ROC analyses for detecting the association between the independent biomarkers associated with time-to-event in the Cox proportional hazards regression analysis and the start of MV are shown in Table 6 and Figure 1. Among the biomarkers at admission, cut-off levels and AUCs were as follows: CRP, 8.0 mg/dL and 0.848; IL-6, 133.0 pg/mL and 0.777; and KL-6, 382.0 IU/mL and 0.707. Similarly, cut-off levels and AUCs among the biomarkers 3 days after admission were as follows: CRP: 1.4 mg/dL and 0.782; HMGB-1, 13.4 ng/mL and 0.857; d-dimer, 1.9 µg/mL and 0.772; and platelet counts, 17.2 cells/µL and 0.704. All biomarkers had AUC values > 0.700 in ROC analyses related to the start of MV. We demonstrated that the ROC analysis results were valid by implementing 5-repeated 10-fold cross-validation (Appendix A). 

Moreover, we analyzed the correlation among biomarkers at admission and 3 days after admission using Spearman’s rank correlation. There was a significant correlation among some biomarkers (Appendix A).

## 4. Discussion

In the present study, we focused on determining independent biomarkers associated with the deterioration of COVID-19 among several candidates reported previously. Moreover, we focused on determining early-phase biomarkers associated with the development of critical respiratory failure requiring MV after admission.

In the present study, we clarified the role of biomarkers that are elevated both immediately and later after tissue damage, such as HMGB-1, a late inflammatory mediator, in COVID-19 patients. HMGB-1 is one of the damage-associated molecular pattern molecules (DAMPs) and endogenous “alarmin” molecules released from dead or damaged cells [24,25,26]. Excessive amounts of extracellular HMGB-1 cause the release of proinflammatory cytokines, including tumor necrosis factor-α, IL-1, IL-18, and IL-6, via the receptor for advanced glycation end products (RAGE) and Toll-like receptor 4. Aberrantly expressed extracellular HMGB-1 is considered to act as a cytokine [27,28]. HMGB-1 is associated with sepsis, malignancy, and immune disease, including ALI/ARDS [29]. In a clinical study, mortality in patients with bacterial pneumonia complicated by ALI/ARDS was strongly predicted by initial appropriate antibiotic use and plasma HMGB-1 levels [30]. HMGB-1 was reported to predict the development of acute exacerbation (AE) appearing pathologically as diffuse alveolar damage, and ALI/ARDS and mortality after AE onset, in idiopathic interstitial pneumonia patients [31,32,33]. Moreover, Chen et al. reported that serum HMGB-1 concentrations are elevated in severe COVID-19 patients, and exogenous HMGB-1 induces the expression of the SARS-CoV-2 entry receptor for angiotensin-converting enzyme-2 in alveolar epithelial cells in a RAGE-dependent manner [8]. Chen et al. also reported that elevated serum levels of S100A8/A9 and HMGB-1 at admission were correlated with high ICU admission rates and mortality in COVID-19 patients [34]. HMGB-1 is a late inflammatory mediator because it first appears in the extracellular environment 8–12 h after the initial macrophage response to proinflammatory stimuli [35]. In this study, the fact that HMGB-1 levels were associated with time-to-MV in serum collected 3 days after admission but not at admission is consistent with this feature. Extracellular HMGB-1 regulates delayed innate immune responses and provides a biomarker for predicting the development of ARDS due to COVID-19.

KL-6, a human MUC1 mucin protein, is a potential biomarker for diagnosing and predicting the severity and mortality of interstitial lung disease [36,37,38,39]. KL-6 concentrations are elevated by injury and/or remodeling of type II pneumocytes and could be detected in serum, pulmonary epithelial lining fluid, and/or bronchoalveolar lavage fluid [33,34,35]. Serum KL-6 concentrations are also elevated in patients with ALI/ARDS compared with controls and in non-survivors compared with survivors [40,41,42]. ARDS is characterized by the influx of protein-rich edema fluid into air spaces because of an increased permeability of the alveolar–capillary barrier. Moreover, pathologically, ARDS patients show fibroblast proliferation and type II alveolar epithelial cell remodeling following acute lung tissue damage [39,43]. Serum KL-6 levels are considered a biomarker for predicting critical respiratory failure in COVID-19 patients by detecting the early phase of ARDS. In some previous reports, the serum KL-6 concentration at admission was higher in severe COVID-19 cases compared with non-severe cases, and the optimal cut-off values were 278.3 to 642.3 U/mL for identifying severe cases [44,45,46,47,48]. Moreover, d’Alessandro et al. reported that KL-6 was higher in COVID-19 patients with fibrotic lung alterations than in the non-fibrotic group [49]. In contrast, Nishida et al. previously reported that elevated KL-6 and surfactant protein D were not significant in pandemic influenza A infections associated with chest radiographic abnormalities. The authors stated that pandemic influenza A infection was likely mainly caused by obstruction of peripheral bronchi, but alveolar involvement was estimated to be minimal [50]. This fact may reflect the difference in the pathophysiology of lower respiratory infection caused by COVID-19 compared with other viruses. 

In the present study, CRP was the only biomarker associated with time-to-MV both at admission and 3 days after admission. In contrast, IL-6 was significantly associated with time-to-MV only at admission. Some studies have reported that IL-6 is an independent biomarker associated with outcomes in COVID-19 patients [9,10]. Popadic et al. reported that IL-6, serum albumin, and d-dimer levels were independent predictors of mortality in patients with moderate to severe ALI/ARDS requiring high-flow oxygen therapy [11]. The present study and previous studies found that serum IL-6 and CRP levels were associated with the need for MV [12]. Hyperinflammation and hypoxia-induced injury caused by SARS-CoV-2 infection induced endothelial cell dysfunction and increased thrombosis and d-dimer [13]. Previous studies showed that d-dimer levels were associated with severity and outcomes in COVID-19 patients [5,11,14,15,16]. Zheng et al. reported that thrombocytopenia, neutrophilia, and lymphocytopenia were independent biomarkers associated with outcomes in COVID-19 patients in a retrospective study, as shown in the present study [51].

The present study has limitations. First, the sample size was small for a comprehensive evaluation of many biomarkers, mainly because the study was conducted at only one center. Second, the study design was retrospective. Third, although the present study focused on determining biomarkers associated with the development of critical respiratory failure by measuring biomarker values early after admission, changes in these values may be associated with the efficacy of therapy and disease progression in COVID-19 patients. Moreover, complications such as diabetes mellitus were able to influence the outcome of study subjects. We plan to analyze the changes in biomarker levels over time, and complications, in a prospective study with larger populations with COVID-19. In addition, the accumulation of studies similar to the present study may enable the prediction of outcomes for this disease. Fourth, biomarkers measured at onset more accurately reflect the clinical course of COVID-19. The lead time bias caused by measuring biomarkers early after admission but not at onset in the present study could not be corrected with a retrospective study design.

## 5. Conclusions

In the present study, we found that increased CRP, IL-6, KL-6, HMGB-1, and d-dimer levels and decreased platelet counts early after admission were associated with the start of MV due to COVID-19. The development of biomarkers that support medical decision making in COVID-19 patients, including whether to initiate MV, is important. Further confirmation is needed to establish the clinical usefulness of these biomarkers because the AUCs were not excellent in the present study. The take-home message of the present study is as follows: “biomarkers that reflect inflammation, lung fibrosis, and coagulation measured at the early phase of the disease were associated with the development of critical respiratory failure due to COVID-19”.

## Figures and Tables

**Figure 1 diagnostics-12-00339-f001:**
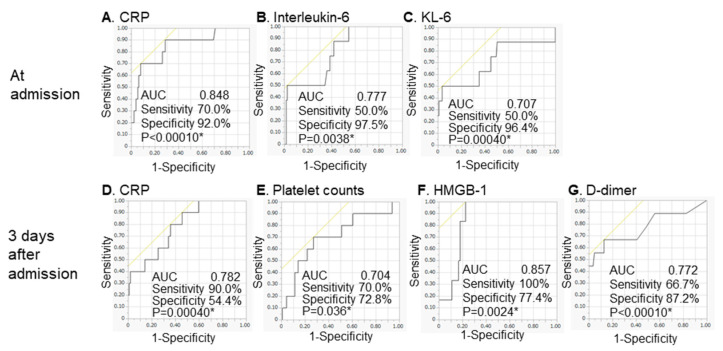
Receiver operating characteristic curves for detecting the association between biomarkers and the start of MV. The receiver operating characteristic curves for detecting the association between independent biomarkers in the Cox proportional hazards regression analysis and the start of MV are shown. (**A**) CRP, (**B**) interleukin-6, and (**C**) KL-6 at admission, and (**D**) CRP, (**E**) platelet counts, (**F**) HMGB-1, and (**G**) d-dimer 3 days after admission. Among the biomarkers at admission, the cut-off levels were as follows: CRP, 8.0 mg/dL; IL-6, 133.0 pg/mL; and KL-6, 382.0 IU/mL. Similarly, the cut-off levels among the biomarkers 3 days after admission were as follows: CRP: 1.4 mg/dL; HMGB-1, 13.4 ng/mL; d-dimer, 1.9 µg/mL; and platelet counts, 17.2 cells/µL. The cut-off levels for the various parameters were defined as the optimal value with the highest Youden index in the receiver operating characteristic curve analysis. COVID-19, coronavirus disease 2019; CRP: C-reactive protein; IL, interleukin; AUC, area under the curve; KL-6, Krebs von den Lungen-6; HMGB-1, high-mobility group box-1 protein; MV, mechanical ventilation. * A value of *p* < 0.05 was considered to represent statistical significance.

**Table 1 diagnostics-12-00339-t001:** Patient characteristics.

	All Cases	Asymptomatic or Mild at Admission	Moderate at Admission	Severe at Admission	*p* Value
*N*	135	42	66	27	
Age (years)	50.0 (35.0–70.0)	32.5 (26.0–49.0)	53.5 (39.0–72.3)	67.0 (51.0–72.0)	<0.00010 *
Gender, Male	69 (51%)	8 (19.1%)	40 (60.6%)	21 (77.8%)	<0.00010 *
Smoker	56 (47.5%)	19 (48.7%)	28 (46.7%)	9 (47.4%)	1.0
Complication					
Diabetes mellitus	22 (16.3%)	1 (2.4%)	11 (16.7%)	10 (37.0%)	0.014 *
Hypertension	33 (24.4%)	4 (9.5%)	16 (24.2%)	13 (48.2%)	0.0014 *
Malignant disease	9 (6.7%)	1 (2.4%)	5 (7.6%)	3 (11.1%)	0.39
Data at admission					
Neutrophil counts (/µL)	3737.0 (2664.0–5883.0)	3278.2 (2303.2–4670.6)	3465.5 (2556.9–5032.9)	5893.0 (4012.0–8333.4)	<0.0001 *
Lymphocyte counts (/µL)	1360.1 (952.0–1757.5)	1635.1 (1038.0–1823.8)	1211.8 (922.1–1610.8)	1354.5 (844.9–1925.7)	0.048 *
Platelet counts (×10^4^/µL)	18.5 (15.6–24.0)	20.4 (16.9–24.5)	18.7 (15.4–25.5)	18.0 (13.6–20.5)	0.18
CRP (mg/dL)	1.5 (0.22–4.7)	0.14 (0.05–0.59)	2.3 (1.1–4.6)	6.9 (3.0–10.7)	<0.00010 *
Lactate dehydrogenase (IU/L)	219.0 (177.0–296.0)	174.5 (156.8–201.3)	220.5 (188.5–264.3)	390.0 (298.0–485.0)	<0.00010 *
Ferritin (ng/mL)	337.0 (114.0–743.1)	99.5 (33.8–136.0)	371.8 (251.6–668.6)	1116.4 (643.0–1698.9)	<0.00010 *
Interleukin-6 (pg/mL)	9.1 (2.4–20.8)	1.8 (1.3–4.5)	11.4 (5.8–21.2)	20.4 (8.4–58.9)	<0.00010 *
Interleukin-18 (pg/mL)	295.0 (207.5–406.0)	227.5 (175.0–310.0)	290.0 (210.0–406.0)	502.5 (353.5–678.8)	<0.00010 *
KL-6 (IU/mL)	220.5 (185.0–294.3)	204.5 (168.8–264.3)	224.0 (184.0–289.0)	312.0 (210.0–410.0)	0.0010 *
HMGB-1 (ng/mL)	6.5 (4.1–9.4)	7.3 (5.8–11.0)	5.4 (3.2–8.8)	6.6 (5.1–11.9)	0.13
Soluble CD163 (ng/mL)	558.5 (459.8–724.0)	513.0 (435.0–694.0)	529.5 (440.5–631.5)	748.0 (501.0–886.0)	0.1
d-dimer (µg/mL)	0.50 (0.30–1.1)	0.40 (0.20–0.55)	0.40 (0.20–1.1)	1.1 (0.60–1.7)	<0.00010 *
Data at day 3					<0.00010 *
Neutrophil counts (/µL)	3011.4 (2073.0–4951.0)	2521.1 (1740.6–3844.9)	2808.0 (2007.8–4048.9)	6706.0 (3063.7–9683.2)	<0.00010 *
Lymphocyte counts (/µL)	1278.9 (900.6–1847.3)	1818.0 (1332.4–2118.0)	1171.8 (792.0–1729.0)	1040.7 (599.8–1321.3)	<0.00010 *
Platelet counts (×10^4^/µL)	20.5 (16.3–26.4)	20.7 (17.0–26.1)	19.8 (15.1–27.2)	22.0 (17.6–28.3)	0.70
CRP (mg/dL)	1.4 (0.26–4.6)	0.070 (0.050–0.32)	2.1 (0.77–7.1)	5.0 (1.8–9.4)	<0.00010 *
Lactate dehydrogenase (IU/L)	214.0 (170.0–306.8)	164.0 (138.0–195.0)	226.0 (181.0–275.0)	371.0 (310.5–491.8)	<0.00010 *
Ferritin (ng/mL)	467.8 (142.3–978.5)	85.1 (21.0–132.2)	469.8 (228.0–964.5)	869.2 (666.6–1681.8)	<0.00010 *
Interleukin-6 (pg/mL)	7.7 (2.4–23.1)	1.8 (1.2–2.7)	10.0 (4.2–20.0)	20.2 (5.8–37.9)	<0.00010 *
HMGB-1 (ng/mL)	8.5 (5.1–14.7)	6.3 (4.3–9.5)	7.0 (5.0–11.0)	15.2 (8.7–21.4)	0.00010 *
Soluble CD163 (ng/mL)	617.0 (459.5–886.0)	463.0 (342.8–630.5)	597.0 (459.3–808.8)	878.0 (650.5–1000.0)	<0.00010 *
d-dimer (µg/mL)	0.60 (0.40–1.3)	0.40 (0.20–0.60)	0.60 (0.40–1.4)	1.0 (0.50–4.8)	0.0016 *

CRP, C-reactive protein; KL-6, Krebs von den Lungen-6; HMGB-1, high-mobility group box-1 protein. * A value of *p* < 0.05 was considered to represent statistical significance.

**Table 2 diagnostics-12-00339-t002:** Patient outcomes.

	All Cases at Admission	Asymptomatic or Mild at Admission	Moderate at Admission	Severe at Admission	*p* Value
*N*	135	42	66	27	
Started MV or died	11 (8.1%)	0	6 (9.1%)	5 (18.5%)	0.015 *
Started MV and survived	6 (4.4%)	0	4 (6.1%)	2 (7.4%)	0.21
Started MV and died	4 (3.0%)	0	1 (1.5%)	3 (11.1%)	0.034 *
Survived without MV	124 (91.9%)	42 (100%)	60 (90.9%)	22 (81.5%)	0.015 *
Died without MV	1 (0.74%)	0	1 (1.5%)	0	1
Duration from admission to start of MV	2.0 (0–8.0)	-	8.0 (1.5–8.5)	0 (0–2.5)	0.084

MV, mechanical ventilation. * A value of *p* < 0.05 was considered to represent statistical significance.

**Table 3 diagnostics-12-00339-t003:** Comparison between patients who started MV and others.

	Started MV		
	Yes	No	*p* Value
*N*	10 (7.4%)	125 (92.6%)	
Data at admission			
Neutrophil counts (/µL)	6104.1 (4162.6–8076.9)	3570.0 (2642.4–5302.8)	0.017 *
Lymphocyte counts (/µL)	1007.0 (749.0–1384.2)	1419.6 (958.0–1765.1)	0.084 *
Platelet counts (×10^4^/µL)	14.4 (12.3–25.9)	18.9 (16.2–23.9)	0.070
CRP (mg/dL)	9.2 (3.7–17.9)	1.4 (0.19–4.2)	0.00030 *
Lactate dehydrogenase (IU/L)	381.0 (229.8–525.0)	213.0 (174.5–277.5)	0.0017 *
Ferritin (ng/mL)	1243.3 (627.1–2095.0)	294.9 (102.5–635.6)	0.00020 *
Interleukin-6 (pg/mL)	315.5 (215.8–1027.0)	213.0 (183.0–285.0)	0.052
Interleukin-18 (pg/mL)	5.0 (3.6–19.1)	6.6 (4.1–9.4)	0.67
KL-6 (IU/mL)	827.5 (541.0–971.5)	539.5 (446.8–695.5)	0.094
HMGB-1 (ng/mL)	7.0 (4.7–11.2)	6.3 (4.0–9.5)	0.67
Soluble CD163 (ng/mL)	717.0 (465.0–886.0)	541.0 (435.0–696.0)	0.19
d-dimer (µg/mL)	1.9 (0.60–3.9)	0.40 (0.20–0.95)	0.00070 *
Data at day 3			
Neutrophil counts (/µL)	6532.0 (3443.0–9246.2)	2953.0 (2044.5–4524.6)	0.012 *
Lymphocyte counts (/µL)	568.3 (440.6–885.5)	1344.0 (983.5–1866.4)	0.00070 *
Platelet counts (×10^4^/µL)	15.3 (13.3–21.8)	20.8 (16.9–26.9)	0.033 *
CRP (mg/dL)	6.0 (2.0–25.2)	1.2 (0.19–4.0)	0.0031 *
Lactate dehydrogenase (IU/L)	326.5 (292.8–446.3)	207.0 (165.8–284.3)	0.0020 *
Ferritin (ng/mL)	1104.4 (640.7–2199.9)	420.5 (126.1–865.9)	0.0022 *
Interleukin-6 (pg/mL)	56.4 (27.9–164.0)	6.7 (2.3–18.6)	0.0011 *
HMGB-1 (ng/mL)	16.4 (14.8–32.9)	7.9 (5.0–11.5)	0.0037 *
Soluble CD163 (ng/mL)	766.0 (502.5–1000.0)	610.5 (459.3–825.8)	0.32
d-dimer (µg/mL)	4.8 (0.55–60.1)	0.50 (0.30–1.0)	0.0071 *

MV, mechanical ventilation; CRP, C-reactive protein; KL-6, Krebs von den Lungen-6; HMGB-1, high-mobility group box-1 protein. * A value of *p* < 0.05 was considered to represent statistical significance.

**Table 4 diagnostics-12-00339-t004:** Univariate analysis by Cox proportional hazards regression analysis for biomarkers associated with time-to-MV.

**A**. Analysis of biomarkers at admission			
	**RR**	**95%CI**	** *p* ** **Value**
Neutrophil (µL)	54.7	3.4–489.5	0.0077 *
Lymphocyte (µL)	0.022	0.00013–1.6	0.086
Platelet (×10^4^/µL)	0.14	0.0014–7.5	0.35
CRP (mg/dL)	51.4	8.7–318.4	<0.00010 *
Lactate dehydrogenase (IU/L)	188.5	14.6–2152.1	0.00020 *
Ferritin (ng/mL)	108.7	10.3–845.6	0.00070 *
Interleukin-6 (pg/mL)	26.6	3.2–139.2	0.0053 *
Interleukin-18 (pg/mL)	7.2	0.057–106.5	0.35
KL-6 (IU/mL)	89.4	8.6–654.1	0.0012 *
HMGB-1 (ng/mL)	1.9	0.0015–55.9	0.81
Soluble CD163 (ng/mL)	10.8	0.34–340.9	0.17
d-dimer (µg/mL)	91.6	9.5–697.5	0.00070 *
**B.** Analysis of biomarkers at 3 days after admission			
	**RR**	**95%CI**	** *p* ** **Value**
Neutrophil (µL)	41.1	2.9–344.9	0.0090 *
Lymphocyte (µL)	0.038	0.00057–0.0018 §	0.33
Platelet (×10^4^/µL)	0.013	0.00012–0.87	0.043 *
CRP (mg/dL)	53.2	7.6–338.6	0.00020 *
Lactate dehydrogenase (IU/L)	25.8	2.5–180.0	0.0087 *
Ferritin (ng/mL)	20.5	2.0–117.0	0.015 *
Interleukin-6 (pg/mL)	45.1	4.7–293.2	0.0031 *
HMGB-1 (ng/mL)	2219.8	33.0–600,284.4	0.00050 *
Soluble CD163 (ng/mL)	3.7	0.29–54.6	0.31
d-dimer (µg/mL)	123.3	10.7–1659.5	0.00070 *

§ parameter estimates. MV, mechanical ventilation; CRP, C-reactive protein; KL-6, Krebs von den Lungen-6; HMGB-1, high-mobility group box-1 protein; RR, relative risk; 95% CI, 95% confidence interval. * A value of *p* < 0.05 was considered to represent statistical significance.

**Table 5 diagnostics-12-00339-t005:** Multivariate analysis by Cox proportional hazards regression analysis for biomarkers associated with time-to-MV.

**A.** Analysis of biomarkers at admission			
	**RR**	**95%CI**	** *p* ** **Value**
CRP (mg/dL)	33.1	3.2–372.6	0.0028 *
Interleukin-6 (pg/mL)	14.5	0.67–142.7	0.041 *
KL-6 (IU/mL)	64.7	3.6–804.4	0.0013 *
**B.** Analysis of biomarkers at 3 days after admission			
	**RR**	**95%CI**	** *p* ** **Value**
Platelet (×10^4^/µL)	−0.37 §	−0.72–−0.14 §	0.00060 *
CRP (mg/dL)	−0.33 §	−0.96–−0.044 §	0.021 *
HMGB-1 (ng/mL)	0.22 §	0.054–0.46 §	0.011 *
d-dimer (µg/mL)	0.93 §	0.083–2.1 §	0.00037 *

§ parameter estimates; CRP, C-reactive protein; KL-6, Krebs von den Lungen-6; HMGB-1, high-mobility group box-1 protein; RR, relative risk; 95% CI, 95% confidence interval. * A value of *p* < 0.05 was considered to represent statistical significance.

**Table 6 diagnostics-12-00339-t006:** Receiver operating characteristic curve analysis for detecting the association between biomarkers and the start of MV.

**A.** Analysis of biomarkers at admission					
	**Cut-Off**	**AUC**	**Sensitivity**	**Specificity**	** *p* ** **Value**
Neutrophil (/µL)	4672.6	0.727	80.0%	69.6%	0.0089 *
Lymphocyte (/µL)	1333.5	0.665	80.0%	54,4%	0.083
Platelet (×10^4^/µL)	14.9	0.672	70.0%	84.8%	0.33
CRP (mg/dL)	8.0	0.848	70.0%	92.0%	<0.0001 *
Lactate dehydrogenase (IU/L)	372.0	0.798	60.0%	89.6%	0.0003 *
Ferritin (ng/mL)	706.5	0.855	80.0%	78.3%	0.0004 *
Interleukin-6 (pg/mL)	133.0	0.777	50.0%	97.5%	0.0038 *
Interleukin-18 (pg/mL)	281.0	0.655	85.7%	50.0%	0.35
KL-6 (IU/mL)	382.0	0.707	50.0%	96.4%	0.0004 *
HMGB-1 (ng/mL)	7.0	0.547	57.1%	39.8%	0.82
Soluble CD163 (ng/mL)	675.0	0.648	71.4%	72.8%	0.19
d-dimer (µg/mL)	1.2	0.789	70.0%	81.3%	0.0006 *
**B.** Analysis of biomarkers at 3 days after admission					
	**Cut-Off**	**AUC**	**Sensitivity**	**Specificity**	** *p* ** **Value**
Neutrophil (µL)	5765.5	0.752	66.7%	83.0%	0.011 *
Lymphocyte (µL)	790.0	0.826	80.0%	85.8%	0.35
Platelet (×10^4^/µL)	17.2	0.704	70.0%	72.8%	0.036 *
CRP (mg/dL)	1.4	0.782	90.0%	54.4%	0.0004 *
Lactate dehydrogenase (IU/L)	308.0	0.795	80.0%	79.2%	0.0078 *
Ferritin (ng/mL)	434.2	0.798	100.0%	50.6%	0.0092 *
Interleukin-6 (pg/mL)	33.8	0.899	83.3%	91.9%	0.0004 *
HMGB-1 (ng/mL)	13.4	0.857	100.0%	77.4%	0.0024 *
Soluble CD163 (ng/mL)	712.0	0.621	66.7%	66.7%	0.32
d-dimer (µg/mL)	1.9	0.772	66.7%	87.2%	<0.0001 *

AUC, area under the curve; CRP, C-reactive protein; KL-6, Krebs von den Lungen-6; HMGB-1, high-mobility group box-1 protein; RR, relative risk; 95% CI, 95% confidence interval. * A value of *p* < 0.05 was considered to represent statistical significance.

## Data Availability

The data presented in this study are available on request from the corresponding author. The data are not publicly available due to ethical considerations.

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
