# Peer review of "Analysis of Early Biomarkers Associated with the Development of Critical Respiratory Failure in Coronavirus Disease 2019 (COVID-19)"

_diagnostics, 2022, doi:10.3390/diagnostics12020339_

Round 1

Reviewer 1 Report

I have received this manuscript for review from Diagnostics; however, I previously reviewed it at JCM. At that time, after taking into account my comments, I proposed accepting the manuscript for publication. At the moment, I can also confirm that the manuscript is worth publishing at its current stage. Previous reviews are available at the editor's request.
I congratulate the authors on their excellent work.

Author Response

Comments and Suggestions for Authors

I have received this manuscript for review from Diagnostics; however, I previously reviewed it at JCM. At that time, after taking into account my comments, I proposed accepting the manuscript for publication. At the moment, I can also confirm that the manuscript is worth publishing at its current stage. Previous reviews are available at the editor's request.

I congratulate the authors on their excellent work.

Response

Thank you for your compliments. We would also like to thank you for accepting review following JCM. I will carefully revise this article by referring to the comments from other reviewer.

Reviewer 2 Report

Authors attempted to clarify whether blood biomarkers measured early after admission are associated with start of MV or death due to COVID-19 infection.

Although this manuscript is potentially interesting, several issues arise.

  1. This study may be small size to evaluate many biomarkers.

  1. The conclusion of abstract is required to be reconsidered. Authors should add the “take home message”.

  1. Table 2. In “Started MV or died”, 6 patients were in moderate illness.

Were the patients with moderate illness generally treated without MV?

  1. Why did the patients with moderate illness die?

 Did they have nay complications?

  1. Why were there no patient with “critical illness”?

  1. Information of comorbidity may be helpful.

  1. Normal values of biomarkers may be helpful.

  1. Table 2. Survivors without MV may be helpful.

  1. Table 6. Why was the cutoff value of CRP low at 3 days after admission?

  1. Suggestions from native speakers may be helpful.

  1. Were there any relationships between several biomarker?

Minor points

The abbreviation of MV in abstract should be shown.

Author Response

Comments and Suggestions for Authors

Authors attempted to clarify whether blood biomarkers measured early after admission are associated with start of MV or death due to COVID-19 infection. Although this manuscript is potentially interesting, several issues arise.

This study may be small size to evaluate many biomarkers.

The conclusion of abstract is required to be reconsidered. Authors should add the “take home message”.

Response

 Thank you for your correct comment. As you pointed, the limitation of this study is small size for the comprehensive evaluation of many biomarkers. We have emphasized this point in the part of limitation as follows. Therefore, we have revised the abstract as follows.

Page 12, line 339, Discussion

The present study has limitations. First, the sample size was small for the comprehensive evaluation of many biomarkers, mainly because the study was conducted at only one center.

Page 1, line 40, Abstract

In conclusion, increased CRP, IL-6, KL-6, HMGB-1, and D-dimer levels and decreased platelet counts were associated with start of mechanical ventilation due to COVID-19.

Page 12, line 359, Conclusion

Take home message of the present study is “biomarkers that reflect inflammation, lung fibrosis, coagulation measured at early phase of disease were associated with development of critical respiratory failure due to COVID-19”.

Comment

  1. Table 2. In “Started MV or died”, 6 patients were in moderate illness. Were the patients with moderate illness generally treated without MV?

Response

 Thank you for your comment. Among the 66 patients with a moderate illness at admission, 5 started MV. Among these 5 patients, 4 survived and 1 died. One of 66 patients with moderate illness died without MV. In the present study, we classified the severity of patients based on the NIH criteria in tables 1 and 2 at the time of admission, because the study focused on determining the biomarker measured early after admission associated with start of MV. I consider that not clearly showing that classification may make the present study confusing. Therefore, we revised the method and table 1 and 2 for clarifying that the severity classification is determined at the time of admission as follows.

Page 3, line 98, Method

This classification of severity was determined by the condition at the time of admission.

Table 2 and 3

 Asymptomatic or Mild → Asymptomatic or Mild at admission

 Moderate → Moderate at admission

 Severe → Severe at admission

Comment

  1. Why did the patients with moderate illness die? Did they have any complications?

Response

 Among 66 patients with moderate illness at admission, 11 (16.7%) have Diabetes Mellitus, 16 (24.2%) have hypertension, 5 (7.6%) have malignant disease. We added information of comorbidity in table 1 and the Result section as follows. We added the limitation section that these complications may influence the outcome of study subjects.

Page 4, line 153, Result, table 1

Among the 135 study cases, 22 (16.3%) had diabetes mellitus (DM), 33 (24.4%) had hypertension, and 9 (6.7%) had malignant disease.

Page 12, line 345, Discussion

Moreover, complications such as Diabetes Mellitus were able to influence the outcome of study subjects. We plan to analyze the changes in biomarker levels over time and complication in the prospective study with larger populations with COVID-19.

Comment

  1. Why were there no patient with “critical illness”?

Response

 Thank you for your comment. The subjects of the present study did not include those with critical illness at admission. The accurate reason of that is unknown. One possible reason could be that a nearby hospital had a ECMO center and actively accepted the patients with critically illness.

Comment

  1. Information of comorbidity may be helpful.

Response

 Thank you for your comment. We added information of comorbidity in table 1 and the Result section as follows. We added the limitation section that these complications may influence the outcome of study subjects. Please refer to the response to comment 2.

Page 4, line 153, Result, table 1

Among the 135 study cases, 22 (16.3%) had diabetes mellitus (DM), 33 (24.4%) had hypertension, and 9 (6.7%) had malignant disease.

Page 12, line 345, Discussion

Moreover, complications such as Diabetes Mellitus may influence the outcome of study subjects. We plan to analyze the changes in biomarker levels over time and complication in the prospective study with larger populations with COVID-19.

Comment

  1. Normal values of biomarkers may be helpful.

Response

 Thank you for your comment. We have added normal value of biomarkers with confirmed that in section of Patients and Methods (Page 3, line 107).

Page 3, line 126, Method

Standard values of biomarkers were as follows: CRP, less than 0.20 mg/dL; KL-6, less than 500 IU/mL; LDH, less than 225 IU/L; IL-6, less than 6.6 pg/mL; soluble CD163, less than 472 pg/mL; neutrophil count, from 1800 to 7500/uL; lymphocyte count, 1000 to 4800 /uL; platelet count range, 15.8 to 34.8×104 /uL.

Comment

  1. Table 2. Survivors without MV may be helpful.

Response

 Thank you for your comment. We have added number of survivors without MV in table 2.

Comment

  1. Table 6. Why was the cutoff value of CRP low at 3 days after admission?

Response

 Thank you for your comment. The accurate reason why median CRP level at day 3 after admission is lower than that at day 3 is unknown. One of the possible reasons is decline by corticosteroid therapy performed in some cases.

Comment

  1. Suggestions from native speakers may be helpful.

Response

 Thank you for your comment. I am sorry for poor English. This original article has been revised by native check. This original article has been proofread in English again. Changes made with reference to the reviewer's comments are highlighted in yellow, while changes made by proofreading in English are highlighted in blue.

Comment

  1. Were there any relationships between several biomarker?

Response

 Thank you for your comment. We performed the multivariate analysis of biomarkers and added the following description and table S2.

Page 9, line 248, Result

Moreover, we analyzed the correlation among biomarkers at admission and 3 days after admission using Spearman's rank correlation. The significant correlation among some biomarkers (table S2).

Minor points

The abbreviation of MV in abstract should be shown.

Response

Thank you for your comment. We revised the abstract as you pointed as follows.

Page 1, line 28, 31, 35, 41

MV → mechanical ventilation

Round 2

Reviewer 2 Report

Authors have sufficiently responded my comments. I have no further comment.